# Pituitary Glycoprotein Hormones in Human Milk before and after Pasteurization or Refrigeration

**DOI:** 10.3390/nu12030687

**Published:** 2020-03-04

**Authors:** Réka A. Vass, Robert D. Roghair, Edward F. Bell, Tarah T. Colaizy, Karen J. Johnson, Mendi L. Schmelzel, Jacky R. Walker, Tibor Ertl

**Affiliations:** 1Departments of Neonatology and Obstetrics & Gynecology, University of Pécs Medical School, 7624 Pécs, Hungary; rekaanna.vass@gmail.com (R.A.V.); tibor.ertl@aok.pte.hu (T.E.); 2MTA-PTE Human Reproduction Scientific Research Group, University of Pécs, 7624 Pécs, Hungary; 3Stead Family Department of Pediatrics, University of Iowa, Iowa City, IA 52242, USA; edward-bell@uiowa.edu (E.F.B.); tarah-colaizy@uiowa.edu (T.T.C.); karen-johnson@uiowa.edu (K.J.J.); mendi-schmelzel@uiowa.edu (M.L.S.); jacky-walker@uiowa.edu (J.R.W.)

**Keywords:** breast milk, diet, preterm milk, donor milk, Holder pasteurization, storage

## Abstract

Our aims were to investigate the presence of pituitary glycoprotein hormones in preterm and donor milk, and to examine the effects of Holder pasteurization and refrigeration on the levels of these hormones. We measured follicle-stimulating hormone (FSH), luteinizing hormone (LH), and thyroid-stimulating hormone (TSH) in milk samples from mothers who delivered prematurely (*n* = 27) and in samples of mothers who delivered at term and donated milk to the Mother’s Milk Bank of Iowa (*n* = 30). The gonadotropins and TSH were present in similar amounts within human milk produced for preterm and term infants. FSH increased 21% after refrigeration (*p* < 0.05), while LH declined by 39% (*p* < 0.05). Holder pasteurization decreased LH by 24% (*p* < 0.05) and increased TSH by 17% (*p* < 0.05). Holder pasteurization followed by refrigeration resulted in a 21% increase in FSH and a 41% decrease in LH (both *p* < 0.05), resulting in more than a 3-fold increase in donor milk FSH:LH ratios (*p* < 0.05 versus fresh donor milk). Despite structural similarities, the gonadotropins are differentially impacted by Holder pasteurization and refrigeration, and this results in marked alterations in the relative amount of FSH and LH that may be administered to preterm infants, potentially swinging hormonal balance towards ovarian hyperstimulation in females and hypogonadism in males.

## 1. Introduction

Human milk is a biologically active fluid that contains hormones with diverse effects on the human infant. After birth, breast milk is the exclusive hormonal link between mother and infant. Due to the disruption in pregnancy, preterm infants are exposed more briefly to maternal hormones than term infants are. Therefore, for those who were born prematurely, hormones from maternal breast milk may play an important role in development. It is known that early breast milk exposure after preterm birth is associated with improved structural connectivity of developing neural networks in the brain [1]. Early endocrinological programming is influenced by the maternal hormones transferred to the offspring. Human milk contains a variety of hormones [2], but the levels of pituitary hormones in breast milk have not been examined.

Follicle-stimulating hormone (FSH), luteinizing hormone (LH), and thyroid-stimulating hormone (TSH) are produced in the anterior lobe of the pituitary gland. During intrauterine development, FSH and LH levels rise to stimulate gonadal development, only to fall to very low levels during the third trimester when transplacental estrogen delivery peaks and suppresses pituitary hormone secretion [3]. Following delivery, estrogen’s effects on responsive tissues wane, and within months of delivery, there is a transient phase of FSH and LH secretion known as mini-puberty [3]. In preclinical models, excessive estrogen exposure during neonatal development can suppress LH production and elicit irreversible infertility [4]. When infants are born prematurely, the lack of transplacental hormone delivery in the context of hypothalamic immaturity places them at high risk of hormonal deficiency. Transient deficiencies of cortisol and thyroxine are well described in the preterm population, and many infants receive replacement therapy with hydrocortisone and/or levothyroxine [5,6]. While we recently detailed the differential content of cortisol in maternal breast milk versus donated breast milk [7], the impact of alternative feeding practices on the delivery of pituitary glycoprotein hormones to preterm infants has not been defined.

When own mother’s milk is not available in the neonatal intensive care unit, donor milk (DM) is recommended for feeding preterm infants [8]. DM is submitted to Holder pasteurization (HoP) to ensure microbiological safety in human milk banks [8,9]. Another everyday breast milk handling practice is refrigeration, which is commonly used to retain the quality of breast milk and to limit bacterial growth in both own mother’s milk and donor milk. The effect of refrigeration on the concentrations of hormones in milk is not known. Our aims were to investigate the presence of FSH, LH, and TSH in preterm and donor milk, and to examine the effects of HoP and refrigeration and the combination of these processes on the levels of these hormones. Through our investigation of these aims, we confirm the presence of pituitary glycoprotein hormones in breast milk and highlight the significant impact that HoP and refrigeration can have on dietary hormone intake.

## 2. Materials and Methods

This study was approved by the institutional review board of the University of Iowa (identification number 201805935). The preterm mothers and donors consented to the use of their milk samples for this study. We recruited 30 donors from the Mother’s Milk Bank of Iowa (MMBI) (Iowa City, IA, USA), who were registered and approved donors and fulfilled the requirements of the MMBI. The mothers were not identified; however, we had access to deidentified data on their age, height, prepregnancy or first trimester weight, infant sex, and the duration of storage of the milk sample. The samples were stored according to the policy of the MMBI by freezing at −20 °C. We also enrolled 27 mothers from the Neonatal Intensive Care Unit of the University of Iowa Children’s Hospital. These mothers had given birth to their children at gestational ages between 24 and 36 weeks. Their samples were stored at −20 °C until measurements were performed. The donor milk samples were analyzed in 3 different forms: the raw, freshly thawed samples, which are presumed to represent the native hormone levels; the pasteurized samples, which showed the effect of pasteurization; and samples that were pasteurized, frozen at −20 °C, then thawed and stored at 4 °C, which is the usual clinical practice. This last group represents samples typically fed to infants in the neonatal intensive care unit (NICU). The preterm milk samples were divided into 2 different groups (raw samples and refrigerated samples) to represent the general practice applied at the hospital and home.

The day before the measurements, we thawed one aliquot of the donor milk samples to room temperature, pasteurized an aliquot for 30 min at 62.5 °C, and then placed an aliquot in a refrigerator at 4 °C for 24 h. Each sample was sonicated to disrupt the milk fat globule membranes and centrifuged at 15,000× *g* for 15 min; then the skim milk was transferred to tubes for the analyses. For the preterm milk samples, one aliquot was first allowed to reach room temperature. We then placed an aliquot in polypropylene tubes in the refrigerator at 4 °C for 24 h. We chose 24 h as the time of refrigeration because in the University of Iowa Children’s Hospital, opened donor milk bottles are allowed to be stored in the refrigerator for up to 24 h. The samples were thawed before the experiment. The samples were sonicated in order to disrupt the membranes of the milk fat globules, and then centrifuged at 15,000× *g* for 15 min at 4 °C. We removed the lipid layer from the top of the samples and used the skim milk layer to perform the assays.

To measure the FSH, LH, and TSH concentrations, we used HPTP1MAG-66K (END Millipore Corporation, Billerica, MA, USA) kits. First, 25 µL of matrix solution was added to the blank, standard, and quality control wells. Then, 25 µL of standard and quality controls and 50 µL of milk samples and later mixed beads solution, were added to all wells on a 96-well plate. After 17 h of incubation at 4 °C, we washed the plates, then added 25 µL of detection antibody followed by 25 µL of Streptavidin-Phycoerythrin to each well. After washing sections, we added 100 µL of sheath fluid to the wells and ran the plates on FLEXMAP 3DTM with xPONENT software. The lower limits for quantification were FSH 10 mIU/L, LH 10 mIU/L, and TSH 10 μIU/L.

For statistical analyses, we used GraphPad (La Jolla, CA, US), and the normality of data distribution was tested using Saphiro–Wilks tests. Statgraphics Centurion XVII, version 17.0.16 (Statpoint Technologies, Inc., Warrenton, VA, USA), and R 3.3.2 (R-project, http://www.r-project.org) software were used to perform the data analyses. Analyte levels below the lower limit of quantification (LLOQ) were assigned values equal to the LLOQ divided by the square root of 2. Differences were considered significant when *p* values were <0.05. The study was powered to detect moderate effect sizes (Cohen’s *d* = 0.6). Data are shown as mean ± SEM or median values with interquartile ranges.

## 3. Results

### 3.1. Maternal Demographics

Demographic data of enrolled mothers showed that the average maternal age did not differ significantly between the 30 Milk Bank donors (30.8 ± 0.8 years) and the 27 recruited mothers (30.3 ± 1.1 years) who gave birth to preterm infants. As anticipated, the gestational age at delivery for the preterm group (30.1 ± 0.8 weeks) was significantly lower (*p* < 0.001) compared to the donor group (39.4 ± 0.2 weeks), in which all mothers delivered at term. The postpartum duration was likewise different, with donor mothers having delivered an average of 5 months (159 ± 13 days) before sample collection versus an average of 4 weeks (28 ± 4 days) among the mothers of currently admitted preterm infants; this difference was statistically significant (*p* < 0.001).

### 3.2. Comparison of Term and Preterm Breast Milk

Of the 57 samples, four had FSH levels below the LLOQ, none had an LH level below the LLOQ, and two had TSH levels below the LLOQ. FSH, LH, and TSH were present in almost the same amounts in preterm and donor milk samples. The median value of FSH in the preterm milk samples was 180 mIU/L (interquartile range [IQR] 75–315 mIU/L), which did not differ significantly from the values detected in the term donor milk (178 mIU/L, IQR 136–241 mIU/L). The median value of LH concentration in the preterm samples was 40 mIU/L (IQR 18–103 mIU/L) and in the donor milk was 50 mIU/L (IQR 31–60 mIU/L). The median value of TSH concentration in the preterm milk was 60 µIU/L (IQR 38–93 µIU/L) and in the donor milk was 50 µIU/L (IQR 36–64 µIU/L).

### 3.3. Effects of Pasteurization and Storage

In milk banks, HoP is the usual method applied to ensure microbiological safety. In our study, by paired analysis, HoP reduced LH levels by 24% (IQR 0%–40%) and increased TSH levels by 17% (IQR −12%–58%), both *p* < 0.05, without significantly influencing the concentration of FSH (Figure 1). We next investigated whether refrigeration, an everyday home and hospital practice for milk storage, influenced breast milk hormonal composition. In a combined cohort of donor and maternal samples, 24 h refrigeration increased the level of FSH by 21% (IQR 0%–35%) and reduced the level of LH by 39% (IQR 17%–62%), both *p* < 0.05, without significantly altering TSH content (Figure 2).

When donor milk was subjected to combined HoP and refrigerated storage, the level of FSH increased by 21% (IQR 8%–34%), reflecting the HoP-independent consequence of prolonged storage (Figure 3). In contrast, the content of LH decreased by 41% (IQR 26%–61%) within those same samples, reflecting the combined effects of HoP and storage (Figure 3). Finally, related to the counterbalancing effects of HoP and storage on TSH level, TSH content in donor milk was unaffected after combined HoP and storage (Figure 3). The contrasting effects of HoP and refrigeration on the two gonadotropins led to the significant increase in FSH:LH ratios that followed HoP plus refrigerated storage (Figure 4).

### 3.4. Exploratory Analysis of Confounding Variables

Exploring potential confounding variables, we found no significant effect of infant gender (preterm samples: *p* = 0.55 for FSH, *p* = 0.28 for LH, *p* = 0.93 for TSH; donor samples: *p* = 0.44 for FSH, *p* = 0.12 for LH, *p* = 0.50 for TSH) or maternal BMI (*p* = 0.31 for FSH, *p* = 0.43 for LH, *p* = 0.82 for TSH) on milk hormone levels (data not shown). Notably, the breast milk sample provided by one NICU mother with a history of undesired infertility had a relatively normal TSH level (105 µIU/L) but extremely high levels of FSH (2760 mIU/L) and LH (1615 mIU/L). The response of that sample to refrigeration (FSH increased by 28% and LH decreased by 19%) was consistent with the effects of refrigeration observed for the overall cohort, and our statistical analysis did not exclude the sample provided by that or any other mother. If it was provided at 150 mL/kg/d, use of that refrigerated milk would lead to a projected FSH and LH intake of 0.53 mIU/g and 0.20 mIU/g, respectively.

## 4. Discussion

Breast milk provides the nutrients needed by the newborn infant, including a myriad of bioactive compounds, such as immunoglobulins, cytokines, growth factors, and hormones [10,11]. Thyroid hormones—controlled by TSH—are necessary for normal growth and control the metabolism of almost every tissue; without them, the cells of the nervous system do not develop and function properly. In infants born before 30 weeks’ gestational age, the TSH and T4 surges are often lacking [12,13]. Therefore, premature infants often experience transient hypothyroxinemia for several weeks due to interruption of the maternal thyroid hormone supply and immaturity of the preterm thyroid axis [12,14]. To our knowledge, we are the first to report the presence of TSH in breast milk expressed for administration to preterm and term infants. The oral bioavailability of TSH is unknown, but even if unabsorbed, the presence of TSH receptors in the intestine raises the potential for local effects [15].

The gonadotropins FSH and LH control reproductive function with indirect impacts on neurodevelopment, and there are important consequences of deficiency during both prenatal and neonatal development [16,17,18]. A marked difference between the sexes is that LH levels exceed FSH levels in male fetuses [19]. The difference between genders is probably caused by the negative feedback that results from the higher concentrations of fetal testicular hormones in males [20,21]. FSH levels are higher in females than in males and peak between 1 week and 3 months. Subsequently, in males the FSH values gradually decrease to the prepubertal range within 4 months of age, whereas in females these values remain elevated until 3–4 years of age [22,23].

We detected both FSH and LH in breast milk provided by mothers of preterm and term infants. We further investigated the effects of HoP and refrigeration on these hormone levels since those techniques are commonly used to process donor milk in milk banks and maternal milk in the neonatal intensive care unit or at home. Compared to fresh breast milk, we found that provision of donor milk that has been subjected to HoP and refrigeration would significantly reduce LH intake, while simultaneously increasing FSH intake. In the infant’s gastrointestinal tract, the gastric digestion of proteins is attenuated during the first 3 months of life, and the cell–cell connections and barrier functions are immature. Therefore, FSH and LH may be absorbed into the circulation and exert systemic effects [24,25]. Oral administration of low-dose recombinant FSH (20 mIU/day) is known to reduce disease severity in a mouse model of polycystic ovary syndrome [26]. Previous investigators have further found that LH can cross the blood–brain barrier [27,28], is present in cerebrospinal fluid [29,30,31] and can be produced by extra-hypothalamic neurons [32].

Two studies have explored the role of infant nutrition on circulating FSH or LH levels, but they have primarily explored the potentially suppressive effect of the phytoestrogens present in soy formula [33,34]. While a comparison of breast milk to formula was not a focus of these investigations, interesting results were noted. In the investigation by Fang and colleagues, there were trends for higher FSH and lower LH levels in boys receiving breast milk rather than either cow formula or soy formula [33]. Likewise, Adgent and colleagues identified significantly faster breast bud regression in breast-fed versus cow formula-fed or soy formula-fed boys [34]. Coupled with our data, these results suggest the need to further assess circulating gonadotrophin levels in bottle-fed male infants receiving pasteurized or refrigerated breast milk. Follow-up investigations assessing adult reproductive outcomes would be a natural extension if such infants are shown to have reduced neonatal LH levels.

In contrast to the predisposition of male infants towards the long-term consequences of hypogonadism, newborn female infants are at risk of excessive pituitary glycoprotein exposure. With impaired negative feedback and immature ovaries, premature female infants are at increased risk of excessive gonadotrophin production and ovarian overstimulation. In extreme cases, this preterm ovarian hyperstimulation syndrome can lead to ovarian torsion [3]. While most cases resolve spontaneously [35], our detection of extremely high FSH and LH levels in the breast milk of a mother with a personal history of infertility suggests that it may be prudent to assess dietary gonadotropin content when preterm ovarian hyperstimulation syndrome is identified. While we had a single sampling point and FSH content may fluctuate, provision of breast milk with the FSH content we identified in that case would be projected to deliver a dose of FSH that approaches the 1 mIU per gram body weight dose that was extrapolated from clinical practice for Tessaro’s investigations [26].

Our study does have limitations. Milk samples were sonicated to disrupt milk fat globules and allow proteins to enter the aqueous phase, but removal of the fat layer prior to analysis may have led to an underestimation of hormone levels. The composition of human milk varies throughout lactation, and the samples that we utilized were collected at only one point in time. While we are unable to isolate time-dependent changes in hormone levels, those factors are unlikely to influence the within-sample effects of HoP or refrigeration. Reflecting the nutritional options typically available for preterm infants, we compared samples of own mother’s milk that were expressed 4 weeks after delivery to samples of milk donated to a milk bank several months after delivery. It is possible that this significant difference in postpartum duration confounded our attempt to detect baseline differences in the content of milk expressed by mothers that delivered preterm rather than term infants.

The mechanisms behind our findings, that the different handling processes differentially affect the levels of these pituitary hormones, are unknown. These glycoprotein hormones consist of two polypeptide units, an alpha unit and a beta unit. Their structures are similar, because the alpha subunits of LH, FSH, and TSH are identical and consist of 96 amino acids, while the beta subunits vary [36]. The applied bioassays are the preferred methods of testing for the biologically active forms of these hormones in breast milk, providing precise information about the effects of everyday milk handling practices, refrigeration and HoP, on hormone levels. Oberkotter and Tenor identified a protein that binds the thyroid hormones in breast milk [37], and others have demonstrated variable effects of HoP on the levels of many other large molecules found in breast milk [38]. We assume that the cold temperature manipulates these bonds and, in this way, may reveal hidden hormone content that is observable only after 24 h of refrigeration, and that heat treatment may cause degradation of some of these hormones. Pasteurized human milk can be stored at −20 °C for up to 8 months without compromising its macronutrient and energy content [39], but little is known about the effects of such storage on the levels of bioactive compounds. The oxidative status of human milk during refrigeration has been examined but with conflicting results. Hanna et al. found that cold storage reduced the antioxidant capacity of milk [40]. Bertino et al. found no such effect [41], similar to the study of Marinković and colleagues, in which the total nonenzymatic antioxidative capacity of the milk was not significantly affected by pasteurization or freezing for a month; however, these breast milk processing techniques caused a significant drop in superoxide dismutase and glutathione peroxidase activity [42].

## 5. Conclusions

Preterm infants suffer from a lack of hormonal impact from the mother’s body. Considering that the third trimester transplacental hormonal influence is missing, the composition and the role of a biologically active fluid like breast milk may be crucial for the developing infant. Our findings demonstrate, for the first time, that three pituitary glycoprotein hormones, FSH, LH, and TSH, are present in the same amounts in preterm and term breast milk, and that everyday milk handling techniques influence the concentration of FSH, LH, and TSH in hormone-specific ways. Further research should explore the implications of an imbalance in the provision of hormones, with attention now drawn to the relatively low intake of LH that follows the HoP and refrigeration of donated breast milk.

## Figures and Tables

**Figure 1 nutrients-12-00687-f001:**
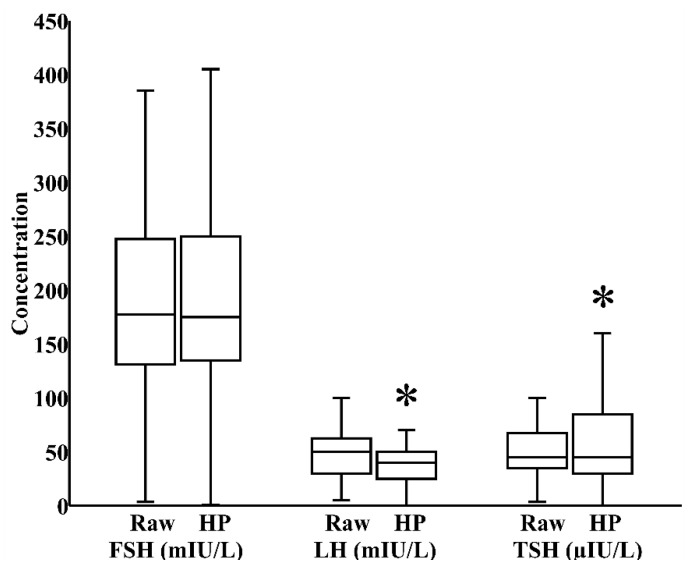
Impact of Holder pasteurization (HoP) on the concentrations of follicle-stimulating hormone (FSH), luteinizing hormone (LH), and thyroid-stimulating hormone (TSH) in breast milk donated to the Mother’s Milk Bank of Iowa (*N* = 30). * *p* < 0.05 versus raw milk.

**Figure 2 nutrients-12-00687-f002:**
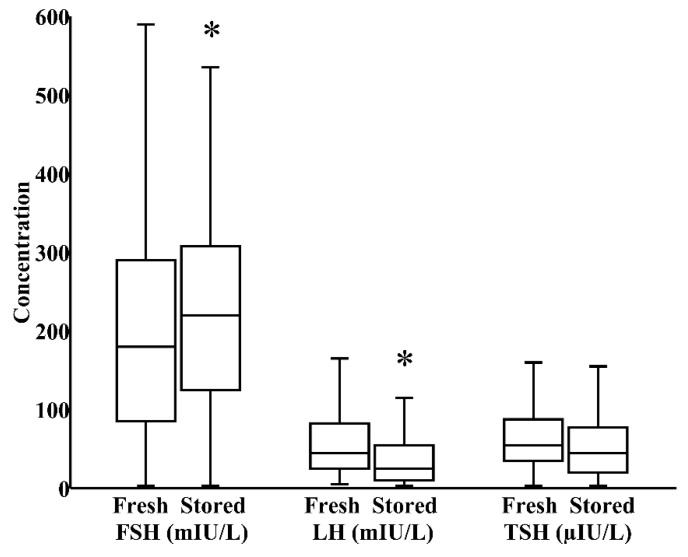
Impact of 24 h refrigerated storage on the concentrations of FSH, LH, and TSH in a combined cohort of freshly pasteurized donor and freshly pumped maternal milk (*N* = 57). * *p* < 0.05 versus fresh milk.

**Figure 3 nutrients-12-00687-f003:**
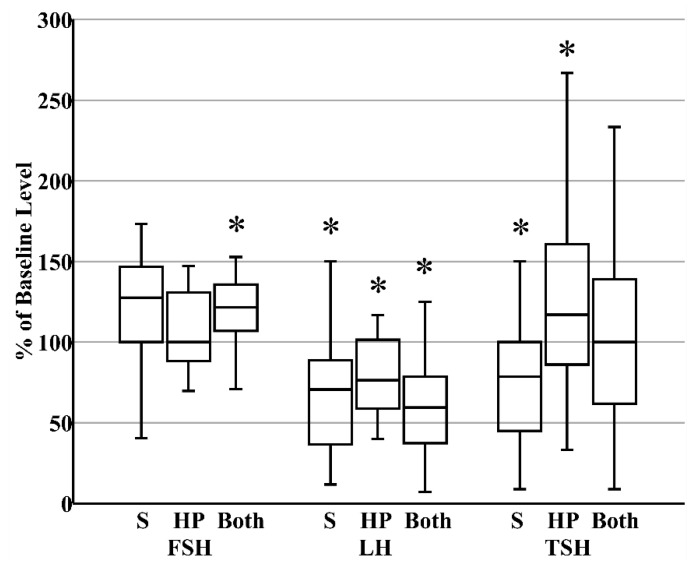
Summary of the changes that occur in the concentrations of pituitary glycoproteins after refrigerated storage (S, *N* = 27, all maternal), holder pasteurization (HP, *N* = 30, all donor), or both storage and holder pasteurization (*N* = 30, all donor). * *p* < 0.05 versus fresh raw milk.

**Figure 4 nutrients-12-00687-f004:**
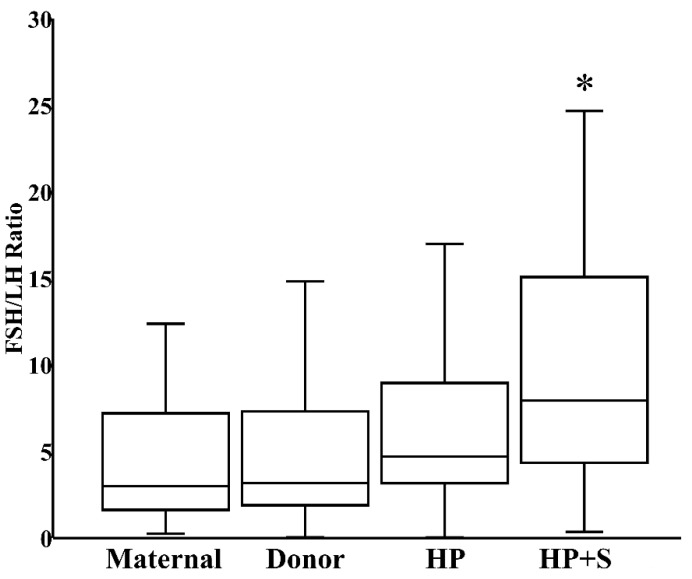
FSH:LH ratio in fresh maternal milk (*N* = 27), fresh donor milk (*N* = 30), pasteurized donor milk (HP, *N* = 30), and pasteurized donor milk after refrigerated storage (HP+S, *N* = 30). * *p* < 0.05 versus fresh donor milk or HoP donor milk.

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
