# Peer review of "Pituitary Glycoprotein Hormones in Human Milk before and after Pasteurization or Refrigeration"

_nutrients, 2020, doi:10.3390/nu12030687_

Round 1
Reviewer 1 Report
The article aims were to investigate the presence of pituitary glycoprotein hormones in preterm and donor milk, and to examine the effects of Holder pasteurization and refrigeration on the levels of these hormones. The authors measured follicle-stimulating hormone (FSH), luteinizing hormone (LH), and thyroid-stimulating hormone (TSH) in milk samples from mothers who delivered prematurely and in samples of mothers who delivered at term. The three pituitary glycoprotein hormones, FSH, LH, and TSH, were present in the same amounts in preterm and term breast milk, and the authors found that everyday milk handling techniques influence the concentration of FSH, LH, and TSH in hormone-specific ways.
General
Maternal milk is a complex bioactive material; do you think that the removal of the milk fat affects the composition of the milk by removing hormones and it could underestimate the hormone levels? Moreover, the human milk composition is different depending on the moment of the day, and on the month of lactation. This could be included in the weaknesses of the study.
Specific Comments:
Introduction: Line 60: “Through our investigation of these aims, we identified the presence of TSH, FSH and LH in term and preterm human milk in concentrations sufficient to raise concern for local and systemic effects. Additionally, HoP and refrigeration differentially altered the levels of the pituitary glycoprotein hormones, significantly decreasing the relative amount of LH that would be provided to infants that receive stored donor rather than fresh maternal breast milk.” This paragraph show results, and it should include in the Results Section, not in introduction/objectives.
Line 115: “The postpartum duration was likewise different, with donor mothers having delivered an average of 5 months (159 ± 13 days) before sample collection versus an average of 4 weeks (28 ± 4 days) among the mothers of currently admitted preterm infants; this difference was statistically significant (p< 0.001).” Do the authors mean that donor’s milk was collected 5 months after delivery and preterm mothers’ milk 4 weeks after delivery? This difference could be considered as a weakness and commented/included in the discussion.
Line 161: “[…] we found no effect of infant gender or maternal BMI on milk hormone levels (data not shown).” This result could include the result of the statistical analysis (p-value).
Line 161: “Notably, the breast milk sample provided by one NICU mother with a history of undesired infertility had a relatively normal TSH level (105 μIU/L) but extremely high levels of FSH (2760 mIU/L) and LH (1615 mIU/L).” Was this mother included or excluded in the statistical analysis?
Author Response
1) Maternal milk is a complex bioactive material; do you think that the removal of the milk fat affects the composition of the milk by removing hormones and it could underestimate the hormone levels? Moreover, the human milk composition is different depending on the moment of the day, and on the month of lactation. This could be included in the weaknesses of the study.
We thank the reviewer for this suggestion, and we added this information to the discussion on lines 224-229. “Our study does have limitations. Milk samples were sonicated to disrupt milk fat globules and allow proteins to enter the aqueous phase, but removal of the fat layer prior to analysis may have led to an underestimation of hormone levels. The composition of human milk varies throughout lactation, and the samples that we utilized were collected at only one point in time. While we are unable to isolate time-dependent changes in hormone levels, those factors are unlikely to influence the within-sample effects of HoP or refrigeration.”
2) The summary of results should be in the Results Section, not in introduction/objectives.
We agree that our attempt to follow the journal guideline and ‘highlight the main conclusions at the end of the introduction’ led to our inclusion of results as well as conclusions. On line 61, we have removed, “we identified the presence of TSH, FSH and LH in term and preterm human milk in concentrations sufficient to raise concern for local and systemic effects. Additionally, HoP and refrigeration differentially altered the levels of the pituitary glycoprotein hormones, significantly decreasing the relative amount of LH that would be provided to infants that receive stored donor rather than fresh maternal breast milk.” In its place, we have added, “we confirm the presence of pituitary glycoprotein hormones in breast milk and highlight the significant impact that HoP and refrigeration can have on dietary hormone intake.”
3) “The postpartum duration was likewise different, with donor mothers having delivered an average of 5 months (159 ± 13 days) before sample collection versus an average of 4 weeks (28 ± 4 days) among the mothers of currently admitted preterm infants; this difference was statistically significant (p< 0.001).” Do the authors mean that donor’s milk was collected 5 months after delivery and preterm mothers’ milk 4 weeks after delivery? This difference could be considered as a weakness and commented/included in the discussion.
We added this point to the limitations on lines 229-234. “Reflecting the nutritional options typically available for preterm infants, we compared samples of own mother’s milk that were expressed 4 weeks after delivery to samples of milk donated to a milk bank several months after delivery. It is possible that this significant difference in postpartum duration confounded our attempt to detect baseline differences in the content of milk expressed by mothers that delivered preterm rather than term infants.”
4) “[…] we found no effect of infant gender or maternal BMI on milk hormone levels (data not shown).” This result could include the result of the statistical analysis (p-value).
We appreciate the suggestion, and p-values have been added to page 5 line 161-163 “…we found no significant effect of infant gender (preterm samples, p = 0.55 for FSH; p = 0.28 for LH; p = 0.93 for TSH; donor samples, p= 0.44 for FSH, p = 0.12 for LH, p = 0.50 for TSH) or maternal BMI (p = 0.31 for FSH, p = 0.43 for LH, p = 0.82 for TSH) on milk hormone levels (data not shown).”
5) “Notably, the breast milk sample provided by one NICU mother with a history of undesired infertility had a relatively normal TSH level (105 µIU/L) but extremely high levels of FSH (2760 mIU/L) and LH (1615 mIU/L).” Was this mother included or excluded in the statistical analysis?
The sentence beginning on line 166 now clarifies, “The response of that sample to refrigeration (FSH increased by 28% and LH decreased by 19%) was consistent with the effects of refrigeration observed for the overall cohort, and our statistical analysis did not exclude the sample provided by that or any other mother.”
Reviewer 2 Report
I thought the paper was well written and brings up an interesting point in regards to the differences between donor milk and pre-term mother's milk.
Author Response
Thank-you for your evaluation of our manuscript and your kind comments.